# Anticipated significant work limitation in primary care consulters with osteoarthritis: a prospective cohort study

Ross Wilkie,[1] Chris Phillipson,[2] Elaine M Hay,[1] Glenn Pransky[3]

## ABSTRACT

**Objective:** To describe the prevalence of expected work limitations (EWL) prior to future retirement age in osteoarthritis consulters, and the associated health, sociodemographic and workplace factors.

**Design:** Population-based prospective cohort study.

**Setting:** General practices in Staffordshire, England.

**Participants:** 297 working adults aged 50–65, who had consulted primary care for osteoarthritis.

**Outcome:** EWL was defined using a single question, "Do you think joint pain will limit your ability to work before you reach 69 years old?"

**Results:** 51 (17.2%) indicated that joint pain would not limit their ability to work until 69, 79 (26.6%) indicated EWL and 167 (56.2%) did not know if joint pain would limit work before 69. In bivariate analysis, physical function (OR 0.93; 95% CI 0.91 to 0.96), depression (4.51; 1.81 to 11.3), cognitive symptom (3.84; 1.81 to 8.18), current smoker (2.75; 1.02 to 7.38), age (0.69; 0.58 to 0.82), physically demanding job (3.18; 1.50 to 6.72), no opportunities to retrain (3.01; 1.29 to 7.05) and work dissatisfaction (3.69; 1.43 to 9.49) were associated with EWL. The final multivariate model included physical function and age.

**Conclusions:** Only one in five osteoarthritis consulters expected that joint pain would not limit their work participation before 69 years of age. Given the expectation for people to work until they are older, the results highlight the increasing need for clinicians to include work participation in their consultation and implement strategies to address work loss/limitation. Targeting pain-related functional limitation and effective communication with employers to manage workplace issues could reduce EWL.

For numbered affiliations see end of article.

**Correspondence to**
Dr Ross Wilkie;
r.wilkie@keele.ac.uk

## INTRODUCTION

Osteoarthritis (OA) is the most common joint condition in adults and globally is the fastest increasing major heath condition.[1] It is a common reason for primary care consultation (1 in 20 consultations in adults aged between 45 and 65 per year is for OA[2]) and is recognised as one of the leading and rapidly growing causes of disability.[3] Its most disabling manifestation (joint pain) is

## Strengths and limitations of this study

- The sample is representative of primary care consulters with physician-diagnosed osteoarthritis.
- The outcome is based on the individual's expectations, but this can be highly predictive of future work loss/limitation and drive consultation for healthcare.
- The methodology enables the prospective identification of clinical, identified by self-report and clinical records, socio-demographic and workplace factors with expected work limitation.

strongly associated with ageing[4] and with the most common forms of disability.[5–9]

Work restriction is one form of disability that will become more important for those with OA and joint pain because increases in state pension age in many developed countries mean that most adults can expect a need to continue working at older ages than before.[10] Normal retirement age in North America and Europe has increased, and is expected to rise further to 69 and beyond.[11] However, the extent to which participation in work will be limited by health-related problems, resulting in significant work limitation in terms of absenteeism and presenteeism (remaining in work but with limitation and reduced productivity), is unclear.[12] The increasing prevalence of chronic health conditions, especially OA, in persons near to retirement age raises questions about the viability of attempts to extend working life. Several studies of expectations of future work loss are predictive of future work outcomes.[13 14] Identifying the prevalence and predictors of expected work limitations (EWL) in this group of patients, particularly those that are amenable to change, will inform management and possible preventative strategies for future work limitation. The aim of this study was to estimate the proportion of working age adults with OA who predict that joint pain would limit their work or stop them

working (ie, EWL) prior to a possible future pension age of 69. In addition, health, sociodemographic or workplace factors associated with EWL, especially those amenable to change, were explored to identify potential targets to manage and prevent EWL.[15–17]

## METHOD

### Study population

The North Staffordshire OA project (NorStOP) is a population-based prospective cohort study. The NorStOP sampling frame comprised all individuals aged 50 years and over who were registered to receive care from one of six general practices in North Staffordshire, England, UK. In 2002, adults aged 50 years and over who gave their written consent for medical record review were followed up over 6 years for consultation to primary care. They were also mailed questionnaires at 3 and 6 years; reminders were sent at 2 and 4 weeks after the initial mailing. The North Staffordshire Local Research Ethics Committee approved this study.

Analyses for this paper included those who (1) consulted for OA during the study period (starting 18 months before the baseline questionnaire was administered, and continuing through the time of the final follow-up questionnaire (ie, from 2000 to 2008)), (2) were of working age (less than 65 years old) and in employment at the 6-year follow-up and (3) completed the item on EWL prior to 69 years of age at 6-year follow-up. Over the study period, there were 923 adults who had consulted for OA and were of working age at 6-year follow-up. Of this group, 398 had retired before state retirement age, 13 were unemployed, 31 were homemakers, leaving 481 who were in employment and thus eligible for the study. Of these, 184 did not complete the item on future work limitation, leaving complete data for 297 participants (adjusted response 61.7%; figure 1). Compared with those individuals who had consulted for OA but did not complete the item on the future work limitation questionnaire (n=184), those included in the analysis (n=297) were more likely to be female (p=0.02) and have an adequate income (p=0.052) but no more likely to be older (p=0.19), have better physical (p=0.91) or mental health (p=0.21) or have gone onto further education (p=0.52).

### Identification of OA

General practitioners in the study used the Read system to code all reasons for clinical encounters in primary care consultations.[18] The Read codes cross-map to the International Classification of Diseases 9/10 (for diseases). Morbidity data (ie, symptoms and diseases) in this system are grouped under 19 main Read chapters. Data collected at the second hierarchical level or above were used to identify diagnostic groups, and these were aggregated starting 18 months before the baseline questionnaire was administered, and continuing through the time of the final follow-up questionnaire. Individuals were

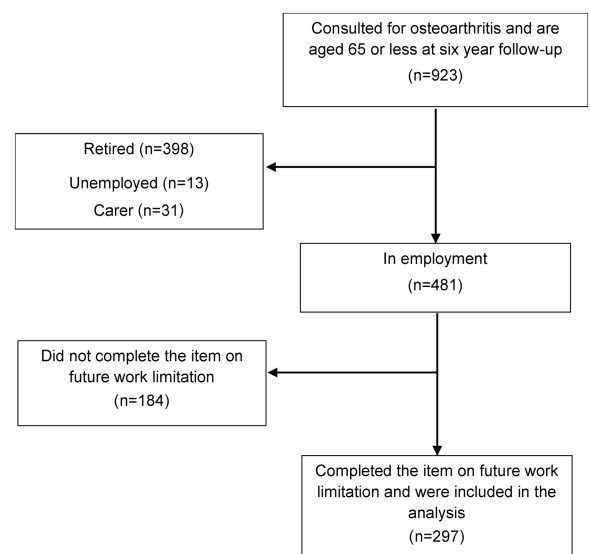

**Figure 1** Flow diagram of participants for the longitudinal analysis.

defined as having OA if they had at least one consultation during this period primarily for OA based on Read codes (N05 category) for primary care consultations.[18] As OA is a long-standing, gradually progressive chronic condition, it was assumed that a clinician-established diagnosis at some point during the study period implied that OA was most likely present, at least to some degree, during the entire period of observation.

### Outcome measure

EWL was defined using a single question at a 6-year follow-up, "Do you think joint pain will limit your ability to work before you reach 69 years old" (will limit or stop me/ don't know/ won't limit).

### Independent factors

Health factors were measured across the 6-year study period, sociodemographic factors at the 3-year follow-up, and workplace factors were measured retrospectively at the 6-year follow-up (table 1).

### Health factors

Physical function was measured at each time point using the physical functioning scale of the Medical Outcomes Study Short Form-36; score range: 0–100, higher scores indicating better function.[19] Items measured the limitation in the individual's capacity to complete basic tasks such as lifting and walking. Scores at each time point were highly correlated (ie, between baseline and the 3-year follow-up r=0.71; between the 3-year and 6-year follow-up r=0.75). Physical function score at the 3-year follow-up (the middle point of data collection) was used in this analysis. The extent of musculoskeletal pain was measured by responders shading painful areas (0–44) on a full body diagram (front and back views). These methods to determine the location and extent of pain

**Table 1** Outcome and independent variables included in the analysis

| | Time point when data were collected | | |
| --- | --- | --- | --- |
| | Baseline | 3-year follow-up | 3-year follow-up |
| **Outcome** | | | |
| Expected work limitation | | | √ |
| **Independent factors** | | | |
| Health factors | | | |
| Physical function | | √ | |
| Extent of pain | √ | √ | √ |
| Comorbidity | √ | √ | √ |
| Smoking | √ | | |
| Depression | √ | √ | √ |
| Anxiety | √ | √ | √ |
| Body mass index | √ | √ | √ |
| Cognitive impairment | √ | √ | √ |
| Control of health | √ | | |
| Sociodemographic | | | |
| Age | √ | | |
| Gender | √ | | |
| Educational attainment | √ | | |
| Occupational classification | √ | | |
| Adequacy of income | √ | | |
| Live alone | √ | | |
| Workplace factors | | | |
| Work type | | | √ |
| Work status | | | √ |
| Physical demands of job | | | √ |
| Flexible working | | | √ |
| Use of aids and appliances | | | √ |
| Opportunities to retrain | | | √ |
| Work satisfaction | | | √ |

are commonly used in population-based studies of pain, and have been shown to be valid and reliable.[20] Using these pain drawings, participants were classified into one of three groups (none, some and widespread). The widespread group were those participants who satisfied the criteria for widespread pain included in the American College of Rheumatology 1990 criteria for fibromyalgia[21] at baseline, 3 years or 6 years. These criteria require pain to be present above and below the waist, on the right and left-hand sides of the body, and in the axial skeleton; the remaining participants who reported pain at any time point that did not satisfy the criteria for widespread pain were classified as having 'some pain' and those participants who did not report pain at all were classified as having 'no pain'.

Comorbidity was identified using Read diagnostic codes from primary care consultations. Multimorbidity was defined as four or more comorbidities (different major diagnostic groups) in the 2 years prior to baseline, between baseline and the 3-year follow-up or between the 3-year and 6-year follow-up.[22] Anxiety and depression during the previous week were measured using the Hospital Anxiety and Depression Scale (HADS)—raw scores categorised individuals as non-cases (0–7) or possible/probable cases (8–21); depression was defined as a

possible or probable case at any of the three time points.[23] Self-reported height and weight were categorised into standard body mass index (BMI) groups (1) normal weight (BMI 20–24.9 kg/m$^2$), (2) underweight (BMI <20 kg/m$^2$), (3) overweight (BMI 25–29.9 kg/m$^2$) and (4) obese (BMI ≥30 kg/m$^2$). Cognitive symptoms were measured using the Alertness Behaviour Subscale of the Sickness Impact Profile.[24] This scale has 10 items that ask about alertness and the ability to concentrate. Each item was scored as 0 (no cognitive symptom) or 1 (cognitive symptom) with raw additive scores categorised to indicate 'no cognitive symptom' (score of 0) and 'cognitive symptom' (score >0); individuals were identified as having a cognitive symptom if they had a score >0 at any of the three time points. Perceived control of health was measured using a single item at baseline (There is a lot I can do to control my health: yes/no). Participants were also asked to report their smoking status at baseline (current, previous or never).

**Sociodemographic factors**

Demographic and socioeconomic details included age, gender, educational attainment (those who finished their education on leaving school; those who went onto

further education such as college or university), occupational class (managerial or professional (chief executive or professor), intermediate (eg, paramedics or technicians), routine (machine operator or childcare worker)), adequacy of income (Thinking about the cost of living as it affects you, which of these descriptions best describes your situation; adequate/inadequate) and living status (live alone/live with others).

## Workplace factors

Single items on the 6-year follow-up questionnaire measured workplace characteristics. Items included work type (Which of the following statements best describes the work that you do in your current job?; sedentary occupation/standing occupation/physical work/heavy manual work), work status (current employment status: full-time/part-time/temporarily off work (eg, due to sickness)), physically demanding employment (Thinking over the past 30 days: Is your work physically demanding?; not physically demanding/physically demanding), flexible working (My hours of work are flexible: flexible/not flexible), coworker support (My work colleagues are supportive: good/low), work satisfaction (How satisfied are you with your current job?; satisfied/dissatisfied), opportunities to retrain (There are opportunities to retrain and develop my skills: yes/no) and able to use aids and appliances and adapt the workplace (I can use aids and appliances to help me do my job or adapt my work: yes/no).

## Statistical analysis

First, the distribution of health, sociodemographic and workplace factors was examined by prediction of the ability to work until age 69 with differences tested for significance using the $\chi^2$ or Kruskal-Wallis tests where appropriate. Regression analyses then focused on identifying factors associated with significant future work loss (will be limited in ability to work, or will be unable to work until age 69 because of joint pain (ie, EWL)), compared with no expected work loss. Bivariate and multivariate logistic regression models were constructed to examine the relationship between each health, sociodemographic and workplace factor and EWL. Following bivariate analysis, multivariate models included all of the factors within each category (ie, all health factors in one model, all sociodemographic in the second model and all of the workplace factors in the third model for each outcome). Factors independently associated with EWL within each category were then included in a final multivariate model. To evaluate the model fit, concordance indexes (C-statistic) were calculated for each model. A C-statistic of 0.50 indicates the predictive ability of a model to be no better than chance, 0.7 indicates reasonable, 0.8 indicates high and 1.0 indicates perfect predictive ability.[25]

Stata V.11 was used for all analyses. The results of the analyses are presented as ORs with 95% CIs. For the regression analyses, the 'won't be limited' group was classified as the referent category.

## RESULTS

Of the 297 consulters for OA included in the analysis, 51 (17.2%) indicated that joint pain would not limit their ability to work until 69 years, 79 (26.6%) indicated that joint pain would limit or stop them working before 69 years (EWL) and 167 (56.2%) indicated that they did not know if they would have EWL before 69 years. Those who indicated EWL (median age 53 years) were younger than those who did not know (median age 54 years) or did not predict EWL (median age 57; p=0.0001; table 2). Women were more likely to indicate that they did not know or would have EWL (p=0.01). There was no significant difference among the three groups for educational attainment (p=0.44) or occupational classification (p=0.10). Notably, all responders with low coworker support predicted developing EWL before age 69 years.

Table 3 shows the results of the regression analysis comparing anticipated EWL to no EWL. In the bivariate analysis, health, socioeconomic and workplace factors were associated with EWL before age 69 years. Of the health factors, depression (OR 4.51; 95% CI 1.81 to 11.3), cognitive symptoms (3.84; 1.81 to 8.18) and being a current smoker (2.75; 1.02 to 7.38) were associated with EWL onset prior to 69 years. Increasing physical function (0.93; 0.91 to 0.96) was protective against EWL onset prior to 69 years. In the multivariate analysis of health factors, only increasing physical function (0.94; 0.90 to 0.97) was associated with EWL onset before 69 years. Of the socioeconomic factors, only age (0.69; 0.58 to 0.82) was associated with EWL. Of the workplace factors, a physically demanding job (3.18; 1.50 to 6.72), no opportunities to retrain (3.01; 1.29 to 7.05) and work dissatisfaction (3.69; 1.43 to 9.49) were associated with EWL. In the multivariate analysis of workplace factors, a physically demanding job and work satisfaction were independently associated with EWL. In the final multivariate model, combining significant factors from the health, socioeconomic and workplace factors, the association with work dissatisfaction attenuated to insignificance (adjusted OR 2.08; 95% CI 0.66 to 6.51). Physical function (0.95; 0.92 to 0.97) and age (0.74; 0.60 to 0.91) remained independently associated with EWL; for every one point increase in physical function, the odds of EWL increased by 7% and for every 1 year increase in age, the odds of EWL decreased by 45%. The model fit of the final model was 0.8418.

## DISCUSSION
### Principal findings

This study is the first to examine the factors associated with an expectation of having significant future work limitation (EWL) for employed patients who consult general practitioners for OA. Physical function, being a current smoker, depression and several workplace

**Table 2**  Participant characteristics overall and by prediction status (n=297)

| | All (n=297) | Will work to 69 without limitation (n=51) | Don't know if joint pain will limit (n=167) | Joint pain will limit or stop me working until 69 (n=79) |
|---|---|---|---|---|
| Age Median (SD) | 54 (2.34) | 57 (2.47) | 54 (2.35) | 53 (2.00) |
| Gender | | | | |
| Male | 134 (45.1) | 32 (60.0) | 64 (38.3) | 38 (48.1) |
| Female | 163 (54.9) | 19 (40.0) | 103 (61.7) | 41 (51.9) |
| Educational attainment | | | | |
| Further | 63 (21.5) | 11 (21.6) | 39 (23.8) | 13 (16.7) |
| School only | 230 (78.5) | 40 (78.4) | 125 (76.2) | 65 (83.3) |
| Live alone | | | | |
| No | 37 (12.9) | 6 (12.2) | 14 (8.6) | 17 (22.7) |
| Yes | 250 (87.1) | 43 (87.8) | 149 (91.4) | 58 (77.3) |
| Occupational classification | | | | |
| Managerial/professional | 73 (24.6) | 17 (33.3) | 40 (24.1) | 16 (20.3) |
| Intermediate | 57 (19.2) | 10 (19.6) | 28 (16.9) | 19 (24.1) |
| Routine | 166 (55.9) | 24 (47.1) | 98 (59.0) | 44 (55.7) |
| Adequacy of income | | | | |
| Adequate | 129 (43.4) | 22 (43.1) | 85 (50.9) | 22 (27.9) |
| Inadequate | 168 (56.6) | 29 (56.9) | 82 (49.1) | 57 (72.2) |
| Physical function Median (SE) | 85 (10.5) | 90 (7.91) | 85 (9.27) | 65 (12.2) |
| Pain status | | | | |
| None | 15 (5.1) | 2 (3.9) | 12 (7.2) | 1 (1.3) |
| Some | 143 (48.2) | 30 (58.8) | 88 (52.7) | 25 (31.7) |
| Widespread | 139 (46.8) | 19 (37.3) | 67 (40.1) | 53 (67.1) |
| Comorbidity | | | | |
| Low comorbidity (0–3) | 122 (41.1) | 20 (39.2) | 78 (46.7) | 24 (30.4) |
| Multimorbidity (4 or more) | 175 (58.9) | 31 (60.8) | 89 (53.3) | 55 (69.6) |
| Depression | | | | |
| Non-case (0–7) | 228 (76.8) | 44 (86.3) | 138 (82.6) | 46 (58.2) |
| Possible/probable case (8–21) | 69 (23.2) | 7 (13.7) | 29 (17.4) | 33 (41) |
| Anxiety | | | | |
| Non-case (0–7) | 121 (40.7) | 27 (52.9) | 65 (38.9) | 29 (36.7) |
| Possible/probable case (8–21) | 176 (59.3) | 24 (47.1) | 102 (61.1) | 50 (63.3) |
| Body mass index | | | | |
| Normal (20–24.9 kg/m$^2$) | 68 (24.7) | 13 (27.7) | 42 (26.4) | 13 (18.8) |
| Underweight (<20 kg/m$^2$) | 9 (3.3) | 1 (2.1) | 6 (3.8) | 2 (2.9) |
| Overweight (25–29.9 kg/m$^2$) | 102 (37.1) | 21 (44.7) | 52 (32.7) | 29 (42.0) |
| Obese (>30 kg/m$^2$) | 96 (34.9) | 12 (25.5) | 59 (37.1) | 25 (36.2) |
| Cognitive symptom | | | | |
| No cognitive symptom | 122 (41.1) | 28 (54.9) | 75 (44.9) | 19 (24.1) |
| Cognitive symptom | 175 (58.9) | 23 (45.1) | 92 (55.1) | 60 (76.0) |
| Smoking | | | | |
| Never | 119 (40.2) | 22 (44.0) | 73 (43.7) | 24 (30.4) |
| Previous | 119 (40.2) | 20 (40.0) | 68 (40.7) | 31 (39.2) |
| Current | 58 (19.6) | 8 (16.0) | 26 (15.6) | 24 (30.4) |
| Control* | | | | |
| Can control health | 277 (93.9) | 48 (94.1) | 157 (95.2) | 72 (91.1) |
| Can't control health | 18 (6.1) | 3 (5.9) | 8 (4.9) | 7 (8.9) |
| Work satisfaction | | | | |
| Satisfied | 198 (74.2) | 43 (86.0) | 115 (75.2) | 40 (62.5) |
| Dissatisfied | 69 (25.8) | 7 (14.0) | 38 (24.8) | 24 (37.5) |
| Work type* | | | | |
| Sedentary | 88 (34.1) | 18 (36.7) | 53 (36.1) | 17 (27.4) |
| Standing | 71 (27.5) | 9 (23.1) | 41 (27.9) | 21 (33.9) |
| Physical | 81 (31.4) | 20 (38.9) | 45 (30.6) | 16 (25.8) |
| Heavy manual | 18 (7.0) | 2 (4.6) | 8 (5.4) | 8 (12.9) |

Continued

| | All (n=297) | Will work to 69 without limitation (n=51) | Don't know if joint pain will limit (n=167) | Joint pain will limit or stop me working until 69 (n=79) |
|---|---|---|---|---|
| **Physical demands of job** | | | | |
| Not physically demanding | 171 (57.6) | 36 (70.6) | 101 (60.5) | 34 (43.0) |
| Physically demanding | 126 (42.4) | 15 (29.4) | 66 (39.5) | 45 (57.0) |
| **Flexible working** | | | | |
| Flexible | 82 (27.6) | 19 (37.3) | 39 (23.4) | 24 (30.4) |
| Not flexible | 215 (72.4) | 32 (62.7) | 128 (76.6) | 55 (69.6) |
| **Use of aids and appliances** | | | | |
| Yes | 151 (50.8) | 32 (62.7) | 79 (47.3) | 40 (50.6) |
| No | 146 (49.2) | 19 (37.3) | 88 (52.7) | 39 (49.4) |
| **Opportunities to retrain** | | | | |
| Yes | 205 (69.0) | 42 (82.4) | 115 (68.9) | 48 (60.8) |
| No | 92 (31.0) | 9 (17.6) | 52 (31.1) | 31 (39.2) |
| **Coworker support** | | | | |
| Good coworker support | 277 (93.3) | 51 (100) | 154 (92.2) | 72 (91.1) |
| Low coworker support | 20 (6.7) | 0 (0) | 13 (7.8) | 7 (8.9) |
| **Work amount** | | | | |
| Full-time | 138 (46.5) | 31 (60.8) | 70 (41.9) | 37 (46.8) |
| Part time | 121 (40.7) | 15 (29.4) | 81 (48.5) | 25 (31.7) |
| Temporary work absence | 38 (12.8) | 5 (9.8) | 16 (9.6) | 15 (21.5) |

**Table 2** Continued

*Missing data.

factors—physically demanding job, work dissatisfaction and poor coworker support—were associated with anticipated EWL. Only one in five OA consulters (17.2%) indicated that joint pain would not limit their ability to work until age 69. Given the high and increasing prevalence of OA, the number of consultations for this disorder and the anticipated extension of working lives due to changes in retirement policy, these results raise significant concerns, while suggesting potential areas for intervention.

The percentage of OA cases who expected work limitation was higher compared to previous studies. Using a national US sample of all workers, Theis et al[15] found that only a third of those with physician-diagnosed arthritis reported work limitations. Unlike the current study, their sample was not restricted to employed persons, and did not ask about the expected ability to work in the future. Studies of patients with rheumatoid arthritis (RA) have found that premature work loss in persons in a similar age group is common, affecting up to 90% of persons with this diagnosis, and leads to significant economic and social consequences.[26] Our finding of a similar high rate of expected work loss in persons with OA suggests that the societal impact in this much larger group will be significant as well.

Physical function was highly protective against EWL. Reduced physical function has been found to be associated with work limitations in persons with any type of arthritis,[15] musculoskeletal disorders in general[27] or those with a specific arthritis diagnosis.[28] [29] This indicates an important mismatch between individual capabilities and work demands, which is more important than

pain by itself. High work physical demands have long been recognised as an important risk factor for subsequent work disability in RA.[27] Some have suggested that control over work demands might be more important than the absolute level of work demands, but these studies did not evaluate the level of job physical demands.[30] Work dissatisfaction was also significant, although this problem was reported in relatively few respondents. There may be several dimensions of work dissatisfaction—not only being dissatisfied at work, but also a preference to be at home instead.[30] Similarly, lack of coworker support was a problem in only a relatively small number of persons, but all who reported low coworker support expected to have EWL. All of those who reported low coworker support also indicated at least one significant health problem (ie, widespread pain, multimorbidity, depression or anxiety). The importance of coworker support in those returning to work after injury or illness, across a range of conditions, has recently been recognised,[31] and low coworker support has been associated with greater job strain and work loss in workers with arthritis.[32] [33] Alternatively, this finding may be less related to coworker support than the particular nature of their job, such as working on their own, or job types that have little involvement with others. If trends in the arrangements of jobs in the future mean that more older adults with OA are working on their own, the significance of low coworker support will increase.

We did not see the effects of low education[15] or comorbidities reported by others. Education may not have a large impact compared with the actual job

**Table 3** Associations between health, demographic, socioeconomic and workplace factors and significant future work limitation in primary care consulters for osteoarthritis, comparing expected work limitation (EWL) with no EWL ORs with 95% CIs

| | OR | 95% CI | Multivariate model within each domain | | Multivariate model including all domains | |
|---|---|---|---|---|---|---|
| | | | OR | 95% CI | OR | 95% CI |
| **Health factors** | | | | | | |
| Physical function | **0.93** | **0.91 to 0.96** | **0.94** | **0.91 to 0.97** | **0.95** | **0.92 to 0.97** |
| Extent of pain | | | | | | |
| No pain | 1 | | 1 | | – | |
| Some | 1.67 | 0.14 to 19.5 | 0.44 | 0.03 to 6.76 | – | |
| Widespread | 5.58 | 0.48 to 65.1 | 0.84 | 0.05 to 13.74 | – | |
| Comorbidity | | | | | | |
| Low comorbidity (0–3) | 1 | | 1 | | – | |
| Multimorbidity (4 or more) | 1.48 | 0.71 to 3.10 | 0.97 | 0.37 to 2.56 | – | |
| Smoking | | | | | | |
| Never | 1 | | 1 | | 1 | |
| Previously | 1.42 | 0.63 to 3.18 | 2.19 | 0.73 to 6.53, | 1.52 | 0.51 to 4.51 |
| Currently | **2.75** | **1.02 to 7.38** | 2.99 | 0.79 to 11.30 | 2.02 | 0.55 to 7.40 |
| Depression | | | | | | |
| Non-case (0–7) | 1 | | 1 | | – | |
| Possible/probable case (8–21) | **4.51** | **1.81 to 11.3** | 1.27 | 0.35 to 4.63 | – | |
| Anxiety | | | | | | |
| Non-case (0–7) | 1 | | 1 | | – | |
| Possible/probable case (8–21) | 1.94 | 0.95 to 3.97 | 1.08 | 0.37 to 3.19 | – | |
| Body mass index | | | | | | |
| Normal (20–24.9 kg/m$^2$) | 1 | | 1 | | – | |
| Underweight (<20 kg/m$^2$) | 1.80 | 0.14 to 23.4 | 1.15 | 0.02 to 64.92 | – | |
| Overweight (25–29.9 kg/m$^2$) | 1.13 | 0.39 to 3.21 | 1.29 | 0.33 to 5.09 | – | |
| Obese (>30 kg/m$^2$) | 2.08 | 0.71 to 6.10 | 1.33 | 0.31 to 5.66 | – | |
| Cognitive impairment | | | | | | |
| No cognitive symptom | 1 | | 1 | | – | |
| Cognitive symptom | **3.84** | **1.81 to 8.18** | 1.98 | 0.70 to 5.58 | – | |
| Control of health | | | | | | |
| Can control health | 1 | | 1 | | – | |
| Can't control health | 1.56 | 0.38 to 6.31 | 1.64 | 0.29 to 9.36 | – | |
| $R^2$ | | | 0.311 | | | |
| C-statistic | | | 0.8641 | | | |
| **Sociodemographic** | | | | | | |
| Age | 0.69 | 0.58 to 0.82 | 0.67 | 0.56 to 0.81 | **0.74** | **0.60 to 0.91** |
| Gender | | | | | | |
| Male | 1 | | 1 | | – | |
| Female | 1.82 | 0.89 to 3.72 | 1.37 | 0.58 to 3.22 | – | |
| Educational attainment | | | | | | |
| Further | 1 | | 1 | | – | |
| School only | 1.38 | 0.56 to 3.36 | 1.12 | 0.34 to 3.64 | – | |
| Occupational classification | | | | | | |
| Managerial/professional | 1 | | 1 | | – | |
| Intermediate | 2.01 | 0.72 to 5.63 | 2.27 | 0.64 to 8.12 | – | |
| Routine | 1.94 | 0.84 to 4.53 | 2.26 | 0.74 to 6.90 | – | |
| Adequacy of income | | | | | | |
| Adequate | 1 | | 1 | | – | |
| Inadequate | 1.97 | 0.94 to 4.12 | 2.01 | 0.84 to 4.81 | – | |
| Live alone | | | | | | |
| No | 1 | | 1 | | – | |
| Yes | 0.48 | 0.17 to 1.31 | 0.50 | 0.16 to 1.56 | – | |
| $R^2$ | | | 0.175 | | | |
| C-statistic | | | 0.7636 | | | |

Continued

**Table 3** Continued

| | OR | 95% CI | Multivariate model within each domain | | Multivariate model including all domains | |
|---|---|---|---|---|---|---|
| | | | OR | 95% CI | OR | 95% CI |
| Workplace factors | | | | | | |
| Work type | | | | | | |
| Sedentary | 1 | | 1 | | – | |
| Standing | 2.47 | 0.89 to 6.88 | 2.32 | 0.60 to 7.92 | – | |
| Physical | 0.85 | 0.33 to 2.15 | 0.46 | 0.13 to 1.59 | – | |
| Heavy manual | 4.24 | 0.79 to 22.8 | 2.38 | 0.30 to 18.7 | – | |
| Work status | | | | | | |
| Full-time | 1 | | 1 | | – | |
| Part time | 1.40 | 0.63 to 3.10 | 1.07 | 0.40 to 2.87 | – | |
| Off work | 2.85 | 0.94 to 8.60 | 0.26 | 0.04 to 1.91 | – | |
| Physical demands of job | | | | | | |
| Not physically demanding | 1 | | 1 | | – | |
| Physically demanding | **3.18** | **1.50 to 6.72** | 1.81 | 0.63 to 5.21 | – | |
| Flexible working | | | | | | |
| Flexible | 1 | | 1 | | – | |
| Not flexible | 1.36 | 0.65 to 2.86 | 1.77 | 0.62 to 5.04 | – | |
| Use of aids and appliances | | | | | | |
| Yes | 1 | | 1 | | – | |
| No | 1.64 | 0.80 to 3.37 | 1.06 | 0.43 to 2.65 | – | |
| Opportunities to retrain | | | | | | |
| Yes | 1 | | 1 | | – | |
| No | **3.01** | **1.29 to 7.05** | 1.99 | 0.72 to 5.45 | – | |
| Work satisfaction | | | | | | |
| Satisfied | 1 | | 1 | | 1 | 0.66 to 6.51 |
| Dissatisfied | **3.69** | **1.43 to 9.49** | **4.78** | **1.57 to 14.6** | 2.08 | |
| R$^2$ | | | 0.116 | | 0.296 | |
| C-statistic | | | 0.7785 | | 0.8418 | |

Results in bold indicate significant association (p<0.05).

physical demands. Flexible work was not a factor, although it has been identified in other studies as a significant predictor of staying at work with RA.[34] This difference may be due to the less specific nature of the question in this study, compared with other investigations.

The majority of participants (52.6%) could not predict whether pain would limit or stop their ability to work until the new retirement age. In additional analysis, a forward stepwise logistic regression model of health, sociodemographic and workplace factors was constructed to identify which factors were associated with being unable to predict whether pain would limit the ability to work. Lower age, being in part-time work and being unable to use aids and appliances were the factors significantly associated with being unable to predict. Their responses may be due to some uncertainty about the future progression of their condition, or about whether or not employment until age 69 would be required. Notably, the greater likelihood of women to indicate 'don't know' was explained by these factors. The role of aids and appliances again indicates the importance of the workplace to allow individuals to self-manage their pain to optimise performance.

**Strengths and weaknesses**

The strength of this study's longitudinal design enables the prospective identification of factors associated with EWL in a clinically relevant primary care population. The sample is representative of primary care consulters with physician-diagnosed OA, relevant to primary care practices. Other studies have been limited to patients from rheumatology practices or rehabilitation clinics, a less representative sample of patients with OA.

There are limitations to this study. A range of factors have been identified as potentially linked to future employment loss in OA, and some were not included in the information collected for this study, such as extent of joint involvement, success of current coping strategies, opportunities for part time work or retirement, importance of work role, social support outside of the workplace, and extent of workplace accommodations specific to their OA condition.[16] In addition, overall pain levels, illness perceptions and health beliefs have not been included, which may have considerable impact on predicted work outcomes.[35] We chose to measure the extent of pain rather than the pain level, as we have previously found extent of pain to be associated with poor work outcomes, and this may be more reflective of total

arthritis impact on physical function than pain intensity.[36] However, clinical measures may not be as important in predicting work disability, compared with measure of function, psychosocial and workplace factors.[34] The C-statistics for the health (0.8641), sociodemographic (0.7636), workplace (0.7785) and final (0.8418) models indicate a reasonable or high ability to predict EWL. The factors included here are those thought to be most important in maintaining employment in chronic musculoskeletal conditions. Data on most variables were by self-report, but validated instruments were used to measure anxiety, depression and pain extent. The outcome is based on individual's expectations, but this can be highly predictive of future work loss/limitation and drive consultation for healthcare.[15–17] We did not have radiographic or detailed information on the extent of OA, but the intention of the study was to describe a typical, heterogeneous group of patients with OA as seen in primary care practice. Measurement of predictors at three time points may not reflect changes in these factors during follow-up. Workplace factors were measured only at 6-year follow-up, and may be out of sequence with health factors, but give a sense of current workplace status at the same time that the question on anticipated EWL was asked. As with any cohort study, non-completion of the items may affect estimates; however. Based on comparisons between those included in the analysis and those who did not complete the EWL item, such effects are likely to be small. The area covered by the study is more deprived on health, education and employment, but with fewer barriers to housing and services, than England as a whole, but again the potential effect of this on estimates will be small.

### Clinical implications

With the current disease prevalence and economic trends, and increasing prevalence of OA, clinicians will undoubtedly have more consultations with persons concerned about their ability to stay at work.[37] Studies in RA suggest that a proactive approach—identification and intervention before work loss has occurred—can be effective in preventing subsequent work loss.[38] In this study, health problems were present well in advance of when EWL was measured. For example, the high correlation between physical function scores at baseline and 3 years (0.71) indicates that low physical function was experienced 6 years before EWL was measured. This indicates that there is a lot of time to intervene with the available strategies.[39] Certain types of coping strategies may be more effective than others in maintaining employment in persons with arthritis, especially anticipating challenges in the workplace and formulating strategies to deal with them in advance.[40] Given the importance of OA-related physical limitations at work as a predictor of EWL, a positive screening question for these limitations should lead to a more in-depth discussion about accommodations and employment options, and perhaps a referral to a vocational expert.[41 42] The

negative association with inadequate coworker support suggests that employer engagement in creating a supportive work environment will also be a major factor in work retention for these patients.[33] Addressing the needs of employed patients with OA within the context of a broader psychosocial model of disease ensures a broader and more relevant perspective on the causes and prevention opportunities for subsequent work loss.[17] There are still many unanswered questions about the optimal nature, timing and duration of interventions designed to maintain employment, and how to engage treating clinicians as positive contributors; thus, further studies will be needed in order to understand how best to address these problems.[43 44]

### CONCLUSION

This observational study suggests that four out of five consulters to primary care with OA expect their joint pain to limit their work participation prior to future pension age, and that younger age and greater arthritis-related physical limitations are the main factors associated with this expected outcome. Given the expectations of people to work until they are older with OA, the results highlight the increasing need for clinicians to include work participation in their consultation and implement strategies to prevent work limitation. Targeting pain-related functional limitation and effective communication with employers to manage workplace issues could reduce the expectation of future work limitation.

**Author affiliations**
[1]Arthritis Research UK Primary Care Centre, Primary Care Sciences, Keele University, Keele, Staffordshire, UK
[2]School of Social Sciences, The University of Manchester, Manchester, UK
[3]Center for Disability Research, Liberty Mutual Research Institute, Hopkinton, Massachusetts, USA

**Acknowledgements** The authors would like to thank the administrative and health informatics staff at Keele University's Arthritis Research UK Primary Care Centre and the doctors and staff of the participating general practices.

**Contributors** RW is the principal investigator of the project, responsible for the conception and design of the study. He was involved in the data analysis and led the drafting and final editing of the paper. CP and EMH are guarantors and advised on the study design. GP is a co-investigator of the project. He led the data analysis and contributed to the drafting and final editing of the manuscript. All authors contributed to the final version of the paper.

**Funding** This study is supported financially by the Medical Research Council, UK (grant code: G9900220) and by the North Staffordshire Primary Care R&D Consortium.

**Competing interests** None.

**Ethics approval** North Staffordshire Local Research Ethics Committee.

**Provenance and peer review** Not commissioned; externally peer reviewed.

**Data sharing statement** The Arthritis Research UK Primary Care Research Centre has established data sharing arrangements to support joint publications and other research collaborations. Applications for access to anonymised data from our research databases are reviewed by the Centre's Data Custodian and Academic Proposal (DCAP) Committee and a decision

regarding access to the data is made subject to the NRES ethical approval first provided for the study and to new analysis being proposed. Further information on our data sharing procedures can be found on the Centre's website (http://www.keele.ac.uk/pchs/publications/datasharingresources/) or by emailing the Centre's data manager (data-sharing-pcs@cphc.keele.ac.uk).

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
