## [Reviewer comments · BMJ Open]

Some articles will have been accepted based in part or entirely on reviews undertaken for other BMJ Group journals. These will be reproduced where possible.

ARTICLE DETAILS

TITLE (PROVISIONAL)	Anticipated significant work limitation in primary care consultants with osteoarthritis: a prospective cohort study
AUTHORS	Wilkie, Ross; Phillipson, Christopher; Hay, Elaine; Pransky, Glenn

VERSION 1 - REVIEW

REVIEWER	Paul Kuijer Coronel Institute of Occupational Health, Academic Medical Center/University of Amsterdam, Amsterdam, The Netherlands.
REVIEW RETURNED	30-May-2014

GENERAL COMMENTS	This is a well written paper about a well performed study on a relevant topic! Hopefully my comments are of use for the authors. Abstract • Informative• No comments Article summary • Well-chosen messages• No further comments Introduction • To-the point• Relevant and up to date references Methods • I found it a bit confusing what parameter is measured at exactly what point in time after inclusion of the participant. Could you please provide a table or figure to get a better overview?• Page 6, line 22, I think that although the p-value=0.052 the group non responders on EWL differ considerably from the responders. I would have added this to the 'more likely to be' instead of 'no more likely to be'. Please address this in the discussion.• Page 8, line 8, Read=READ• Page 8, line 24, ... and weight was (or were?)...• Please use one phrase for 'demographic and socio-economic' throughout the paper. Now it differs for instance 'socio-demographic' or 'socio-economic'.• Page 9, line 3: please also provide an example for 'routine'• Page 10, line 13: I find it informative to get information about the explained variance of the multivariate regression models. Is that manageable? Results
---

- Page 10, line 34-38, add 'years' after 69
- Page 10, line 45, add 'significant' before '... difference among the three...'
- Page 11, line 5, 2.72=2.75
- Page 11, line 22-22, please add OR and 95%CI for 'physically demanding job and work satisfaction' in the text.
- Page 11, line 31-32: could you explain in the result or in the discussion what the OR mean for age and physical function for instance for participants in different age and physical function groups. This provides in my opinion more clinical relevance for a physician that sees a patient with a certain age and physical function score.

Discussion

- Relevant topics are discussed
- Page 13, line 22: could you be more specific than 'If this changes in the future the implications of this will increase.'
- Page 13, line 50: delete one of the '.' after 'ability to work..'
- Page 14, line 12-15 '... current workplace status at the same time that question on anticipated EWL was asked and page 14, line 43-46, '... and workplace problems were present well in advance of when EWL was measured'. Are these two statements in line? See also my former suggestion about 'I found it a bit confusing what parameter is measured at exactly what point in time after inclusion of the participant. Could you please provide a table or figure to get a better overview?'
- Page 16, line 10: The two recent papers of Kievit AJ et al. about a specific questionnaire (Work, Osteoarthritis or joint-Replacement Questionnaire - WORQ) to assess problems performing knee demanding work in patients with osteoarthritis of the knee might be a nice example of a recent development in screening:
 - o Kievit AJ, Kuijer PPFM, Kievit RA, Sierveelt IN, Blankevoort L, Frings-Dresen MHW, A Reliable, Valid and Responsive Questionnaire to Score the Impact of Knee Complaints on Work Following Total Knee Arthroplasty: The WORQ. J ARTHROPLASTY 2014;ahead of print [PubMed]
 - o Kievit AJ, van Geenen RCI, Kuijer PPFM, Pahlplatz TMJ, Blankevoort L, Schafoth MU, Total Knee Arthroplasty and the Unforeseen Impact on Return to Work: A Cross-Sectional Multicenter Survey. J ARTHROPLASTY 2014;ahead of print [PubMed]

Conclusion

- Page 16, Address also the second aim of the study about factors that predict EWL, see for instance your conclusion in the abstract.

Reference

- 27. Demons=persons

Table 1

- Age: I prefer standard deviation instead of standard error
- Some percentages do not count up to 100.0% for instance
 - o Occupational classification ('All' and 'Joint pain will limited or stop...')
 - o Cognitive complaint ('All')
 - o Control ('Don't know...')
 - o Work type ('Will work to...')
 - o Flexible working ('Will work to...' and 'Don't know...')
 - o Use of aids and appliances ('Will work to...')
 - o Opportunities to retrain ('Will work to...')

	 • Table 2  o Is it possible to make the significant ORs more visible for instance printing them bold or using symbols? o As said, is it informative to present also the explained variance? • Figure 1  o No comment Again, an informative and well written paper on a topic that needs more attention of professionals in primary and secondary care. Hopefully my comments are clear and of use for the authors.
--	--

REVIEWER	Carol Coole University of Nottingham UK
REVIEW RETURNED	30-Jun-2014

GENERAL COMMENTS	Thank you for inviting me to review this paper, which reads well and is a subject of importance to a range of stakeholders. I recommend it should be published subject to some very minor changes. Work loss/limitation: sometimes the term ‘work loss’ is used alone in the text when referring to the outcome measure, rather than ‘work loss/limitations’. Method The method of measuring pain seems to be measuring the site of pain rather than the degree of pain – the latter might have a different association with physical function. Illness perception and beliefs have not been measured, and there are no references to these factors. which could have a considerable impact on EWL – and be an area to be addressed with patients, GPs and employers in the management of osteoarthritis in the workplace. Independent factors – could the authors clarify what is meant by ‘health factors being measured across the six year study period’, also ‘workplace factors were measured retrospectively’. Health factors – if these were being measured across the six year study period, could the authors explain why the three year follow up score was used, rather than the six year score. Occupational classification – it would help the reader to have an explanation of ‘intermediate’ and ‘routine’ occupations. Method/Results What was the extent of missing data for other measures, and how was this dealt with? Discussion Second paragraph. The reference to Theis and Murphy implies that the current study has measured reported work limitations. The claim made in the introductory sentence isn’t supported. Suggest revising the paragraph.
---

	Third paragraph – last sentence – suggest the authors clarify what 'this' refers to each time.
--	--

VERSION 1 – AUTHOR RESPONSE

Reviewer: Paul Kuijer

Methods

1. Could you please provide a table or figure to get a better overview of when things were measured?
Response: We have included a table which outlines when the outcome and potential predictors were measured (please see Table 1). We have also added a sentence in the methods explaining that the identification of consulters with osteoarthritis occurred across the study period.

2. Page 6, line 22, I think that although the p-value=0.052 the group non responders on EWL differ considerably from the responders. I would have added this to the 'more likely to be' instead of 'no more likely to be'. Please address this in the discussion.

Response: We agree the initial interpretation is based on the empirical cut-off of 0.05 for significance. We have amended the manuscript (revised manuscript with track changes page 6, para 1). We have alluded to this in the discussion; despite responders being more likely to have an adequate income compared to those who dropped out, income adequacy, or any of the other socio-economic variables were significantly associated with EWL in this study. We expect the level of non-response bias would be minimal.

3. Text amendments: a. Page 8, line 8, Read=READ, b. Page 8, line 24, ... and weight was (or were?)... c. Please use one phrase for 'demographic and socio-economic' throughout the paper, d. Page 9, line 3: please also provide an example for 'routine'

Response: We have revised the text accordingly

4. Page 10, line 13: I find it informative to get information about the explained variance of the multivariate regression models. Is that manageable?

Response: We have included the R2 values for each multivariate analysis (Table 3)

5. Results: Text amendments a. Page 10, line 34-38, add 'years' after 69, b. Page 10, line 45, add 'significant' before '... difference among the three...' c. Page 11, line 5, 2.72=2.75, d. Page 11, line 22-22, please add OR and 95%CI for 'physically demanding job and work satisfaction' in the text, e. Page 13, line 50: delete one of the '.' after 'ability to work..', f. Reference, 27. Demons=persons

Response: We have revised the text accordingly.

6. Page 11, line 31-32: could you explain in the result or in the discussion what the OR mean for age and physical function for instance for participants in different age and physical function groups.

Response: We have added a sentence to explain the result (page 11, para 1).

7. Page 13, line 22: could you be more specific than 'If this changes in the future the implications of this will increase.'

Response: We have revised this sentence (page 13, para 1).

8. Page 14, line 12-15 '... current workplace status at the same time that question on anticipated EWL was asked and page 14, line 43-46, '... and workplace problems were present well in advance of when EWL was measured'. Are these two statements in line?

Response: The questions on EWL and workplace factors were all measured at 6 years. We agree that some of the workplace factors may not have been present well in advance of this time point and have removed reference to the workplace in this sentence.

9. Page 16, line 10: The two recent papers of Kievit AJ et al. about a specific questionnaire (Work, Osteoarthritis or joint-Replacement Questionnaire - WORQ) to assess problems performing knee demanding work in patients with osteoarthritis of the knee might be a nice example of a recent development in screening:

Response: We have added these references to the paper (page 15 para 2)

10. Conclusion: Page 16, Address also the second aim of the study about factors that predict EWL, see for instance your conclusion in the abstract.

Response: We have added additional text to emphasize the results for the predictive factors (page 17, para 2).

10. Table 1 amendments, Age: I prefer standard deviation instead of standard error, Some percentages do not count up to 100.0% for instance.

Table 2: Make the significant ORs more visible for instance printing them in bold. Add explained variance

Response: We have amended the tables accordingly. Please note some percentages do go over 100.0% due to rounding up of decimal places.

Reviewer: Carol Coole

1. Work loss/limitation: sometimes the term 'work loss' is used alone in the text when referring to the outcome measure, rather than 'work loss/limitations'.

Response: We have revised the paper to ensure that work loss/limitation has been applied consistently.

2. The method of measuring pain seems to be measuring the site of pain rather than the degree of pain – the latter might have a different association with physical function.

Response: We included extent of pain because in previous work we found that number of pain sites was associated with poor work outcomes (Wilkie et al. J Occup Rehabil. 2013 Jun;23(2):180-8) and will reflect the total arthritis impact on physical function (page 15, para 1). We agree that pain intensity would correlate highly with physical function; including both would result in over adjustment in the multivariate model. We have added to the text explaining our selection of pain variable (page 15, para 1).

3. Illness perception and beliefs have not been measured, and there are no references to these factors.

Response: We have added to our discussion of limitations to highlight this (Page 14, para 3)

4. Clarify when factors were measured

Response: We have added Table 1 to clarify when factors were measured.

5. Health factors – if these were being measured across the six year study period, could the authors explain why the three year follow up score was used, rather than the six year score.

Response: We used the score at three years because it was the midpoint in our data collection and scores at the three time points were strongly correlated. We have added to the text to explain this (page 6, para 2).

6. Occupational classification

Response: We have provided examples for all classifications of occupational class (page 8, paragraph 2).

7. Method/Results

Response: What was the extent of missing data for other measures, and how was this dealt with?

Response: There were only 2 variables with missing data; control (n=2) and work type (n=39) (please refer to Table1). Whilst the level of missing data for work type can be considered as a limitation, work type was not associated with EWL nor with any of the other workplace variables. In addition we ran the multivariate analysis for workplace factors without worktype in the model and there was no change in the significance of the association of any variables. The extent of missing data was not sufficient to impact on the conclusion of this study.

8. Discussion: Second paragraph. The reference to Theis and Murphy implies that the current study has measured reported work limitations. The claim made in the introductory sentence isn't supported. Suggest revising the paragraph.

Response: We have amended the discussion accordingly; we have clarified that the link between reduced physical function and other work outcomes (page 12, para 3).

9. Third paragraph – last sentence – suggest the authors clarify what 'this' refers to each time.

Response: We have revised this sentence accordingly (page 13, para 1).

We trust that these revisions are satisfactory. We look forward to receiving your response to our re-submission in due course.

VERSION 2 – REVIEW

REVIEWER	Carol Coole University of Nottingham UK
REVIEW RETURNED	22-Jul-2014

GENERAL COMMENTS	The minor revisions I recommended have been addressed and I look forward to seeing the paper published.
---